# Perceived Consequences of Extended Social Isolation on Mental Well-Being: Narratives from Indonesian University Students during the COVID-19 Pandemic

**DOI:** 10.3390/ijerph181910489

**Published:** 2021-10-06

**Authors:** Maila D. H. Rahiem, Steven Eric Krauss, Robin Ersing

**Affiliations:** 1Faculty of Education, UIN Syarif Hidayatullah, Jakarta 15412, Indonesia; mailadinia@uinjkt.ac.id; 2Institute for Social Science Studies (IPSAS), Universiti Putra Malaysia, Selangor 43400, Malaysia; 3Faculty of Educational Studies, Universiti Putra Malaysia, Selangor 43400, Malaysia; 4School of Public Affairs, University of South Florida, Tempa, FL 33620, USA; rersing@usf.edu

**Keywords:** adolescents, mental health, COVID-19 pandemic, online learning, social distancing, social emotional support

## Abstract

Despite several recent studies reporting on young people’s well-being during COVID-19, few large-scale qualitative studies have been carried out that capture the experiences of young people from low- and middle-income countries (LMICs) undergoing extended social restrictions. The challenges faced by young people from LMICs during COVID-19 are likely to be amplified by their countries’ large populations, resource constraints, lack of access to health care, living conditions, socio-spatial contexts, and the pandemic’s ramifications for communities. This study explored how youths perceived their well-being after being isolated for one-and-a-half years during the COVID-19 pandemic. Qualitative narrative research was employed as a method of inquiry. One-hundred and sixty-six university students in Jakarta, Indonesia, between the ages of 17 and 22 wrote reflective online essays on the consequences of extended pandemic isolation on their mental health. This data collection strategy offered an in-depth understanding of the phenomenon through the narratives of those who experienced it. Seven themes expressing the youths’ perceived well-being were identified through inductive reflective thematic analysis: (1) the anguish of loneliness and estrangement; (2) a state of “brokenness” resulting from emotional agony and distress; (3) frustration, confusion, and anger; (4) the experience of conflicting emotions; (5) uncertainty about both the present and future; (6) a sense of purpose and fulfillment; and (7) turning to faith. The findings provide important insights into Indonesian youths’ well-being following extended social restrictions following the outbreak. Their collective experiences can be used to inform policy and practice regarding the nature of support mechanisms required both during and following the pandemic, and in the future if such a situation were to occur again.

## 1. Introduction

The World Health Organization (WHO) classified the 2019 coronavirus (COVID-19) outbreak as a global pandemic on March 11, 2020 and containing the spread of the virus has been an international priority ever since [1]. Many governments worldwide have introduced social restrictions to isolate their respective populations within their houses, commonly referred to as “quarantine” and “social distance,” to decrease the community spread of the pathogen [2]. These policies impose restrictions on outdoor activities, leading to detrimental impacts on quality of life and mental health both immediately and over time [3]. According to a recent nationwide sample of American adults, COVID-19 stay-at-home orders are associated with depression, generalized anxiety disorder (GAD), insomnia, and acute stress, whereas social distancing behavior is associated with depression, GAD, intrusive thoughts, and stress [4].

The risk factors for mental health are related to social isolation, confinement, loss of freedom, and a restriction of autonomy, as highlighted by various studies [5,6,7]. The WHO has expressed concern about the pandemic’s impact on mental health and psychosocial well-being in the forms of loneliness, anxiety, melancholy, insomnia, extensive alcohol and drug use as well as self-harm or suicidal behavior, which are all believed to be on the rise [8]. While people typically experience anxiety and a sense of uneasiness when their environment changes, increased media exposure, financial stress, and limited capacity to manage task duties have also contributed to negative mental health outcomes during the COVID-19 pandemic [9].

Earlier epidemic-related mental health studies have confirmed a link between young people’s heightened sensitivity and less effective coping strategies compared to older adults [10]. Numerous investigations have found that younger persons are more susceptible to the effects of the COVID-19 pandemic [11,12]. In a recent study involving a national sample of Canadian parents (n = 1472), Moore et al. [13] reported that children and youth are at an increased risk of developing a variety of collateral psychological problems as a result of restrictions on their movement and play activities. Meanwhile, Orgilés et al. [14], who investigated the emotional impact of quarantine on Italian and Spanish children and adolescents aged three to eighteen years old, found that 85.7 percent of parents reported that their children’s emotional conditions and behaviors changed over the course of confinement. The most often reported symptoms were difficulties concentrating (77%), boredom (52%), hostility (39%), restlessness (38%), nervousness (38%), feelings of loneliness (31%), discomfort (30%), and anxiety (30%).

While studies are still scant compared to those from wealthier nations, COVID-19’s impact on children and youth in low- and middle-income countries (LMICs) may be substantially more severe. Young people in LMICs are significantly more vulnerable to the pandemic due to their living conditions, socio-spatial contexts, and the pandemic’s impact on critical social support networks including caregivers, families, peers, and communities. Overcrowded living conditions and lack of clean water in many LMICs make it nearly impossible to follow the necessary public health guidelines aimed at mitigating the spread of the virus [15]. The tremendous economic strain exacerbated by COVID-19 in LMICs has resulted in an increase in school dropouts and child marriages, both of which are frequent strategies employed by disadvantaged families to alleviate financial hardship [16]. Students in LMICs regularly face barriers to accessing online technology due to unequal digital landscapes that prevent youth from accessing the Internet and related virtual spaces [17,18]. Furthermore, schools are frequently a primary source of food for children in many LMICs [19]. School food programs are critical for maintaining adequate nutrition, which is tied to academic achievement and cognitive development [20]. Closure of schools because of the pandemic has increased the risk of food insecurity among school-age children and youth, which has an overall negative effect on their mental health [21].

Due to a lack of adequate health care and social safety nets, sizable proportions of LMIC populations have experienced mental health problems during the COVID-19 pandemic [22]. Approximately one out of four individuals in LMICs have reported significant levels of anxiety. This is likely attributable to being exposed to COVID-19 or being stigmatized as a result of COVID-19 exposure [23]. Examining data from previous pandemics in sub-Saharan Africa along with current data from COVID-19, researchers studying the pandemic in Ghana, Nigeria, and Kenya have uncovered evidence of potential short- and long-term impacts on adolescent health, educational, and psychological well-being [24]. These findings mirrored studies among youth cohorts in Ethiopia, Peru, India, Nepal, and Vietnam, where stress and mental health problems have increased as a result of COVID-19 [18,25]. Studies on the mental health effects of college-going youth in LMICs remain scant. One study reported the prevalence of mild-to-severe symptoms of depression, anxiety, and stress among Bangladeshi university students during the pandemic [26]. The triggers were financial insecurity, fear of COVID-19 infection, a lack of valid information about COVID-19, and excessive exposure to COVID-19 news on social and mass media [27].

Limited past research on the COVID-19 pandemic experiences of university students from LMIC contexts has prompted us to investigate the pandemic’s impact on undergraduate students’ well-being. In general, university students have higher rates of anxiety, depression, substance misuse, and disordered eating than the general population; their mental health burden tends to increase during profound life changes [13,14,28] Elmer et al. [29] examined the university students’ social networks and mental health prior to and during the COVID-19 pandemic. The authors found that student stress levels, anxiety, loneliness, and depressive symptoms worsened during the crisis compared to pre-crisis levels. Aristovnik et al. [30] examined the experiences of university students in 62 nations, collecting 30,383 responses to an online questionnaire. They discovered that throughout the pandemic, students expressed concern about their academic and professional futures as well as boredom, anxiety, and frustration. Browning et al. [28] collected over 2500 survey responses from students at seven colleges in the United States to assess the psychological effects of COVID-19. Their study indicated that COVID-19 had a largely negative influence on the students’ psychological health and lifestyle habits including a lack of motivation, anxiety, stress, isolation, social withdrawal, educational changes, and decreased social interaction.

### 1.1. The Current Study

Research on university student well-being during COVID-19 has been dominated by quantitative survey research designs, with several studies employing secondary data with parents. Few have engaged in qualitative, narrative-based inquiry that captures the experiences of the students in their own words. This literature has also been dominated by studies from high income countries (HICs). The current study used qualitative narrative research to elicit first-hand insights and observations from Indonesian university students about the effects of pandemic isolation on their mental well-being. Students were asked to write reflective essays about how were functioning during the pandemic including their emotions, feelings, and reactions related to being under extended lockdown. The researchers provided a brief explanation of the study, inviting students to openly share their views about being confined to their homes since the start of the lockdown (since March 2020). This approach provided the students with a platform to discuss everything from personal challenges to interpersonal interactions, their concerns, hopes, and any academic difficulties they had encountered.

### 1.2. Study Context

Indonesia is a developing economy with a population of 270.2 million. The COVID-19 pandemic has dimmed prospects for achieving the nation’s development goals. According to the Central Statistics Agency [31], Indonesia’s unemployment rate reached 8.75 million people in February 2021 and is predicted to rise, increasing by 26.26 percent from 6.93 million people during the same period last year. The health crisis resulting from the pandemic has resulted in a decline in Indonesia’s per capita income from USD 4050 in 2019 to USD 3870 in 2020 [32]. It has wreaked havoc on the country’s essential quality of life, due primarily to a lack of resources and a heavily indebted poor population [33]. The pandemic has harmed the country’s economic growth, employment, and social welfare [32], and has lowered life expectancy by up to 1.7 years [34]. It has also affected food production, which is a significant setback given the country’s continued reliance on agriculture [35]. As a consequence of the pandemic, Indonesia was recently reclassified as a Low Middle-Income Country (LMIC) only a year after it was classified as an Upper Middle-Income Country (UMIC) in July 2020 [36].

The impact of the pandemic on Indonesian youth is already being felt in several ways. The pandemic has exacerbated the youth unemployment rate in the country. Pre-pandemic, Indonesia had a considerably higher rate of youth unemployment than the global average. By 2020, the rate had reached nearly 20%—the highest among Southeast Asian countries [37]. The pandemic has also aggravated working conditions for young people by increasing layoffs and shorter working hours [38]. The contribution of young people to nation-building is also contingent on their level of competence and capacity as human resources. Education is a critical precursor to a country’s economic success. Most of Indonesia’s youth (74.18%) are graduates of high school and junior high school [39].

Statewide school closures began in March, 2020. As a result of the pandemic, educational institutions have modified their course delivery methods. The absence of conventional in-class instruction disproportionately affects low-income students [40] who have more difficulty accessing and navigating online learning platforms, preventing them from obtaining high-quality education. The pandemic has also negatively impacted Indonesia’s overall school enrollment rate [41]. World Bank researchers estimate that Indonesian children have lost 11 points on the PISA reading scale and USD 249 in future annual individual earnings as a result of the four-month closure period from March to July, 2020 [42]. These losses are expected to be even more significant as most schools have remained closed as of September 2021, the time of writing this article.

Against this background, the study findings can be used to inform practitioners and policy makers on appropriate interventions for students in higher education. Indonesia’s example can serve as a lesson for other LMICs experiencing extended lockdowns due to the COVID-19 pandemic. Inadequate efforts to identify and address youth mental health issues may have long-term consequences for the students’ health and education [28]. Effectively responding to these issues and supporting youth will help foster national development goals.

## 2. Materials and Methods

### 2.1. Study Design and Participants

The purpose of the study was to examine the effects of pandemic isolation on youth, with a focus on their mental health. We used a narrative inquiry approach to structure the essay question and to analyze the data [43]. Narrative inquiry allows researchers to explore individual experiences and the social, cultural, and institutional narratives that shape, convey, and transmit those experiences [44]. Narratives assist audiences in understanding how individuals perceive complex surroundings [45]. Narrative research also effectively elicits individual perceptions of unusual events and crises [46] including how individuals make sense of catastrophes and their aftermath [47].

Due to restrictions in accessing students, we selected participants using a nonprobability, homogenous sampling technique. The sample consisted of 166 university undergraduates attending a public university in Jakarta, where the first author was based and had established networks to recruit participants. Sixty-one percent of the students who participated were female, with a mean age of 19 years. Ninety-five percent were living on Java island, primarily in the greater Jakarta area. The sample consisted of 41% second-year students, 39.5% third-year students, and 19.5% fourth-year students. All of the study participants were from either early childhood education (54%) or social science education (46%) (see Figure 1).

The sample size was relatively large for a qualitative study. The reasons for this are threefold. First, the main research topic of the students’ mental health is broad in scope and objective—the broader the scope and objective, the higher the sample size needed [47,48]. Second, the sample criteria were less precise—the less precise the participant characteristics are regarding the study objectives, the greater the sample size required [49]. Third, less was known about young Indonesian’s perceptions of pandemic isolation and mental well-being—the less developed the underlying theory, the larger the sample size needed [49]. Additionally, because the primary data collecting technique was reflective essays, a larger sample size allowed for the compensation of insufficient data in the essays submitted by the students. However, the number of students who actually submitted essays exceeded our expectations.

### 2.2. Procedures

The primary study data consisted of reflective essays on the perceived effects of pandemic isolation on the students’ mental health. Students were invited to write about how they felt and functioned during the pandemic. The essays were collected online through Google Form. In the form, we outlined the scope of the research describing the study goals and ensured the confidentiality of the participants’ identities and responses. Participants were informed that participation was voluntary, and that they were allowed to withdraw from the study at any time. We encouraged the participants to write essays of any length. Submitted essays ranged from 400 to 1000 words. The data collection period was 1–15 June 2021 (see Figure 2).

Following collection and analysis of the essays, we also conducted two virtual focus group discussions (FGD) using Zoom to gain a more holistic view of the participants’ experiences and to conduct member checking of the essay responses. We invited 35 participants to partake in the FGD sessions—those who we judged to have submitted the most descriptive, insightful, and well-written essays. Eighteen students participated in the first FGD session, while 17 participated in the second. During the first meeting, we invited participants to add stories or insights that they had not previously shared. The second meeting was used as a member check, during which we shared the analysis summary and solicited feedback on whether any information needed to be revised, removed, or added. Member checking helped to improve the study’s accuracy, credibility, and transferability. This is consistent with Birt et al. [50]’s practice of member checking, in which data or results are displayed to the participants for the verification of accuracy and resonance with their expressed experiences. To procure ethical clearance for the research, the research design and data collection protocols were submitted to, and a letter of authorization for data collection was released by, the first author’s university Center for Research and Publications.

### 2.3. Data Analysis

We employed two unique but complementary analytical approaches for examining the data: thematic and narrative. Thematic analysis is advantageous for examining multiple examples and identifying commonalities and contrasts across a dataset [51], whereas narrative analysis effectively analyzes particularity and situates it in more general contexts [52]. Thematic analysis began on a semantic level to identify surface meanings [51] whereas narrative analysis focused on the specific linguistic mechanisms employed in the stories that were told [53].

To carry out the analysis, we input all 166 essays into a single Microsoft Excel document. Essays were given an identifying code according to the student’s program, year of study, and number in the series. Each essay was then carefully read and re-read by the researchers. During the re-reading phase, we selected stories that met specific criteria, keeping in mind the study objective of exploring the effects of pandemic isolation and mental well-being. Here, the purpose was to explore the content of the narratives and to identify reoccurring themes in each story. Finally, we grouped the initial conceptual categories within each narrative. We coded each narrative separately and then reconfigured and reanalyzed the data from the initial coding procedure, categorizing codes that were the same or comparable. Following the observations of patterns in the classified codes, we arrived at summative conclusions in the form of themes, which became the research findings.

Using an inductive approach, we identified 267 codes in total, which were then classified into 14 negative and 11 positive categories (see Figure 3 and Figure 4). We did not analyze the frequency with which codes appeared and how many codes were in each category. However, frequencies were used to assist in determining the similarities and differences between the participants’ perspectives. We then examined the patterns revealed by the categorized data. We analyzed and contrasted categories and combined those that were similar into themes. We used a Microsoft Excel spreadsheet to organize and carry out all phases of the analysis. We kept all of the data for the first cycle of analysis in a single sheet: the participant’s acronym in one cell, the participant’s essays adjacent to it, and the codes following. We prepared a fresh sheet for the second coding cycle and used the “analyze it” option to consecutively aggregate similar codes. We manually sorted the list in order to construct categories by grouping related and comparable codes together. Categories were subsequently evaluated for patterns and then themes (see Figure 5).

We ensured rigor of the methods employed and trustworthiness of the analysis [54] by employing a homogeneous sampling approach [55], triangulation, and member checks. Triangulation was carried out through the combination of essay writing and FGDs. During the second FGD, member checking was conducted by presenting the analysis and findings to the participants, which were then adjusted following the discussion and participant recommendations. The combination of strategies provided adequate opportunity for the study participants to contribute their perspectives and experiences. We used non-biased and non-leading questions by asking participants to express their feelings in the essay and encouraged additional comments during the FGD sessions.

## 3. Results

From the analysis, seven themes emerged describing how the students perceived their well-being after 1.5 years of being isolated during the COVID-19 pandemic: (1) the anguish of loneliness and estrangement; (2) a state of “brokenness”—emotional agony and distress; (3) frustration, confusion, and anger; (4) the experience of conflicting emotions; (5) uncertainty about current and future events; (6) a sense of purpose and fulfillment; and (7) turning to faith. To elucidate each theme, we included excerpts from both the essays and student remarks during the FGDs. The essay data were coded with ESSAY following the participant’s accounts, and the FGD data were coded with FGD.

### 3.1. The Anguish of Loneliness and Estrangement

The students discussed how pandemic isolation generated deep feelings of loneliness, isolation, and disconnection from others. These feelings often resulted in intense emotional responses. Having no one to talk to, P4A21 stated that she cried and hugged herself for hours at a time, sometimes until she was exhausted, constantly reassuring herself that everything would be alright. “I don’t have anyone,” she commented (P4A21/ESSAY). The comparable sense of loneliness that caused several students to become unexpectedly sorrowful and cry on occasion was also described by P4B32, who wrote about feeling alone and uncared for despite living with her parents and three sisters. “I don’t know why I feel alone” (P4B32/ESSAY). The loneliness that P4A21 and P4B32 felt was experienced even while in the presence of family, reflecting the estrangement that many students felt due to the extended time away from others outside the home.

While some had difficulty articulating the source of their distress, others were clearer as to the cause of their intense loneliness—their inability to meet friends. During the FGD session, M4A27 expressed a desire to see and spend time with his mates. He muttered that a year was an awfully long time apart from friends. He maintained contact with his classmates through online learning, but the time spent was far less in quantity and quality. He lacked meaningful friendships. “I miss seeing my friends and spending time with them… I’m feeling disconnected from my friends and the outside world” (M4A27/FGD).

Similar sentiments were expressed by students who lived far away from home, especially international students who shared how difficult it was to be separated from their families. For example, M4A1, a Turkish international student, missed her family in Turkey, especially after discovering that the virus had afflicted her mother and two siblings and after she was also hospitalized due to illness. “I miss my family and my homeland terribly. I miss my siblings, my mother’s cooking, my dad, and my home’s warm and caring atmosphere” (M4A1/ESSAY).

### 3.2. A State of “Brokenness”—Emotional Agony and Distress

The pupils described feelings of sadness and distress during the lockdown. They were pained by the loss of family members due to COVID-19. During the FGD, M4C22 shared that she lost her father, cousin, and aunt all in a short period of time due to COVID-19. Not being able to help them receive treatment quickly enough due to financial constraints made the event even more traumatic for her and caused much grief every time she recalled it. “It’s agonizing. It still hurts, and I can feel it” (M4C22/FGD).

Several participants described their state as sensitive, melancholy, and even disturbed by the situation, adding that they cried frequently. P4C5 wrote that she had become exceedingly emotional and cried easily. She remarked that she was in tears while writing the essay, but that being prompted to write about her feelings helped her to feel a sense of support. She was quickly moved by the kindness shown to her during these trying times. “I became a crybaby, sobbing uncontrollably… Even now, I am in tears” (P4C5/ESSAY).

The students described their distress further in relation to the financial difficulties they and their families had encountered during the lockdown. M4C3 wrote about his family’s financial situation being more precarious due to his brother being laid off from work. His parents were in their eighties. His family struggled to put food on the table, and when they did, they had to walk to the forest to harvest leaves to eat. He felt backed into a corner, which made him want to surrender. “Please don’t force us to give up, Allah,” he said (M4C3/ESSAY).

### 3.3. Frustration, Confusion, and Anger

In addition to severe sadness and distress, emotions of frustration, confusion, and even anger were shared by the students as a result of having to study and spend most of their time at home. Family conflict was a major reason for the students’ feelings of frustration. While some expressed sadness and distress over family-related issues, others described their feelings as frustration by what was happening in their families, particularly the ongoing feuding between their parents and other family members throughout confinement. M4A30 said that his parents’ relationship problems worsened during the pandemic, prompting them to divorce, and he was frustrated by the situation. “I’m feeling quite depressed, even frustrated…particularly during the days after my mother and father became divorced” (M4A30/ESSAY). He revealed that the primary source of their parents’ conflict was economic difficulties. However, the divorce resolved no issues. It only added to his frustration. “My family’s economic situation has suffered a significant downturn. All of this leaves me frustrated as I don’t know what to do” (M4A30/ESSAY).

Numerous students voiced their confusion regarding online learning during the pandemic. They were overwhelmed by the assignments, much more so when the lecturer did not adequately explain the assigned tasks. “I’m sick of remote learning; I can’t bear it due to the countless assignments assigned by the professor. This confuses and stresses me out” (P4C25/FGD). Still on the matter of studying, due to their dissatisfaction with online learning, the youth shared their concerns about their future, of not fulfilling their parents’ hopes. “I am pretty concerned about my future, mainly because I am my parents’ first hope. How can I graduate from college if I don’t understand what I’m studying in class? I cry every night because I’m so confused” (P4C16/ESSAY). Their apprehension about the future also caused several participants to have difficulties sleeping, “it’s becoming increasingly difficult to sleep on time” (P4B5/FGD). In severe cases, their anxiety over the future led to suicidal thoughts, which they wrote about, “I’m confused about my future, about financing for university, and my family situation. I started overthinking everything. I had considered suicide since I thought I had nothing to fight for” (M4B21/ESSAY).

The students also expressed feelings of rage in describing their experience during isolation. Some students said that they were prone to explode in a fury over trivial matters, and it was not easy for them to sustain a positive mood. “I exploded uncontrollably, enraged even. I am fearful of my rage. I’m scared this will be a lasting shift that I will be powerless to reverse in the future” (P4B31/ESSAY). They expressed anger over not being able to study online effectively due to poor Internet connectivity and domestic noise. “I’m mad; I’m unable to concentrate on online classes due to signal issues; on top of that, it’s extremely noisy at home; when will this madness end?” (M4B15/ESSAY). They alluded to the need for a break from the mayhem, and felt the need to express their anger, “What I’m feeling at the moment is the need for a break from this crazy situation. I’m not sure what I feel or want to feel. I wish I had a way to express my wrath. I need a break” (P4B24/ESSAY).

### 3.4. The Experience of Conflicting Emotions

Not all narratives were straightforward or depicted a single strand of emotions. Several students described how they had experienced diverse feelings, at times conflicting, during the pandemic. Some even mentioned how much they enjoyed being forced to stay at home. P4C5 indicated that he felt “great” and was even pleased to have spent most of the last year and a half at home. He emphasized, however, that at times, this was accompanied by negative emotions. “The feeling is mixed and at times contradictory. Staying at home for more than a year due to this pandemic is incredible. However, numerous emotions exist, including happiness, tiredness, sadness, and anger” (P4C5/ESSAY). One of the reasons for their mixed emotions included the tension of being able to spend as much time as possible with their families, but also losing a sense of privacy. They also added that spending too much time together provoked tensions. “I enjoy spending time with my family, although it may be exhausting at times. I miss being alone, having my own personal space in my rented room. When we spent too much time together, the squabbles began” (M4A29/ESSAY).

Students also had conflicting sentiments about online learning. At times, they enjoyed it, but loathed it at other times. They discovered that studying at home was ineffective due to the lack of imposed schedules with defined time restrictions. Lecturers occasionally shifted online meeting times at the last minute and assigned too much work, forcing students to stay up all night to complete their assignments. In contrast, it also allowed for additional freedoms that they did not have under normal face-to-face learning circumstances. “I am not a fan of online learning. The lecturers overburdened us with homework that was time-intensive and had no time limits. They might just change the time of the class meetings. Though I am kind of happy to eat a snack during class and not have to get up early to ride my bike to campus” (MPB34).

Several students reported experiencing mood swings or abrupt shifts in their emotional states. In a matter of seconds, a happy emotion could morph into a negative one, and vice versa. “During the COVID-19 pandemic, my mood is easily influenced, especially when I’m isolating myself; I can feel happy, alone, and even despondent at times” (P4C16/ESSAY). Furthermore, others felt that not having someone to confide in, typically close friends, resulted in mood fluctuations. They were unaware of how others felt without in-person encounters, which caused them to overthink and could cause their emotions to shift unexpectedly. “I usually discussed anything with my best friend. I haven’t seen her in over a year and a half. It is very different. I think about everything way too much. Sometimes I just want to cry when I was laughing the minute before” (M4B29/ESSAY).

### 3.5. Uncertainty about Current and Future Events

The students felt as though life was getting increasingly uncertain and the situation was spiraling out of control. They were confused, wondering why the pandemic took so long to end and when it would be over. “I’m despondent. We have hope that this pandemic will end soon, but there are still many positive cases, and we literally have no idea when it will end” (M4B11/FGD). They questioned whether their lives would return to normal as they had known them before the outbreak, and they yearned for it. “I miss life without fear, traveling, and seeing my friends. We are very unsure about everything right now. Will we live as we did before the pandemic?” (P4B12/ESSAY).

They questioned why schools remained closed while malls, stores, and restaurants remained open (in Jakarta, where the university is located, educational institutions were closed from March 16, 2020 through to the end of August 2021). They feared the pandemic would never cease and would even get worse. “I was truly unhappy because, even though malls and marketplaces had opened, schools remained closed. It’s been happening for far too long now, and I’m concerned the situation will worsen” (P4C30/ESSAY). The students desired an end to the pandemic to resume in-person classes. They perceived that the quality of online learning was insufficient and poor. “How long do you expect this situation to last? Perhaps remote learning is the best way to study right now, but it’s a long way off when it comes to quality. When asked about concentration, it may be somewhat limited” (P4C30/ESSAY).

The students worried about their future, fearful of how long the pandemic would endure, how they would complete their end-of-study projects, and how they would find employment later. “Everything is unpredictable. It will be difficult to complete my studies and projects, and after I graduate, will I be able to find work” (P4A15). They became more uncertain when the delta variant infection cases began to increase at the end of July 2021. The situation, which appeared to be under control, became concerning as the number of infections and deaths increased, hospital occupancy reached capacity, and the government imposed emergency restrictions on community activities. “I expected it to end soon. Suddenly, many people in my circles became infected, and death news was frequent. Our movements were restricted by the authorities, and many public locations were shuttered” (M4B11). However, several students stated that the COVID-19 situation taught them that the future is uncertain, which caused them to be more deliberate in their acts. “Nothing is certain in this world, as COVID-19 shows us. The pandemic altered the order of things. This makes us believe that the future is unpredictable, so we must be highly vigilant in our actions” (P4C8/ESSAY).

### 3.6. A Sense of Purpose and Fulfillment

Despite the crisis and resulting volatility, the students shared moments of optimism, mostly through an underlying sense of purpose. Several students stated that they discovered a deeper sense of meaning in their lives and felt fulfilled by doing good deeds for others. P4B14 enjoyed assisting her widowed aunt with the latter’s business. “I feel fulfilled to be able to assist my widowed aunt in selling food in the market from dawn until 8 a.m. It is also reciprocal because my aunt then offers free breakfast for my parents, me, and my sister” (P4B14/ESSAY). Mental health is often supported by having a sense of responsibility to others. Another student wrote about the importance of being involved in activities that offered structured activities and goals, especially those directly related to the pandemic. “I have goals in life because I have responsibilities. Along with several other students, I am a member of the district COVID-19 Task Force, which aims to educate the public on the importance of health protocols” (P4B25/FGD).

Several students returned to their villages and established new routines. They wrote that they had to adjust to the new conditions they faced and tried to develop new habits in response. “We must be able to live alongside this virus. Right now, I’m content with my current situation. I returned to my hometown, adjusted to a new study routine, and altered my everyday activities” (P4B23/ESSAY). The students indicated that this helped them have a sense of optimism for the future, believing that the situation would improve. It did not happen by itself, however, but by establishing goals for their lives and working toward them. “I will always try to survive under whatever conditions to accomplish my objectives and dreams. Everything will get better; we are living through it all. I have plans and will work to achieve them” (M4C11/ESSAY).

### 3.7. Turning to Faith

Living in a predominantly Muslim country where religion plays a major role in everyday life, it was not surprising to hear students share that their faith helped them to cope with the difficulties they were facing during the lockdown. In their essays, many expressed gratitude to God, despite the negative feelings brought about by the pandemic. They reported spending more time attempting to draw closer to God through prayer and trusted in God’s wisdom for having to go through such a difficult experience. P4C1 spent much time reflecting on her life and speaking with God, through which she found solace. “I’ve come to appreciate God’s blessings on my health, to reflect on myself, to converse with God and nature, and to contemplate everything that has transpired. I’ve found peace” (P4C1/ESSAY).

They were grateful, recognizing their circumstances as a gift from God and counting themselves fortunate to be healthy. “I am grateful because, despite the pandemic, my family and I have kept our bodily and spiritual health. Many people have died due to COVID-19, and I am happy that I was able to continue my studies” (M4C23/ESSAY). They were grateful they could still eat during a time when their family’s economic situation deteriorated. “Thank God, we still have food on the table; my dad lost his job, and now only my mom works. Our lives have changed, but we should be grateful” (M4B7).

Some students were afflicted, or had family members who tested positive, with COVID-19. To face this test and cope with the possibility of severe illness or even death, they drew strength from their faith. COVID-19 infected M4C10 and her family. Her mother, elder sister, brother-in-law, younger sister, and nephew were all infected. Despite her hardship, she stated that she had gained numerous lessons, particularly patience in coping with illness. Her mother and sister spent nearly three weeks in the hospital due to comorbidities, and she was terrified. She took peace in prayer and believed that it was their prayers that kept them alive. She was relieved when they were ultimately discharged from the hospital, and her entire family tested negative. “I prayed and worshipped more. Thank God, my family and I have been pronounced free of COVID-19 as a result of prayers, and support from neighbors and relatives” (M4C10/FGD).

## 4. Discussion

The purpose of this study was to explore how Indonesian university students perceived their state of well-being after being isolated for one and a half years due to the COVID-19 pandemic. We did this by analyzing 166 reflective essays and carrying out two FGDs with 35 students. The study findings showed that the youth perceived extended pandemic isolation as exacerbating feelings of being alone and disconnected. Young adults appear to be the most susceptible to experiences of loneliness in comparison to other age groups [56]. Recent studies on youth during COVID-19 have reported that more than one-third of adolescents and nearly half of 18–24-year-olds have experienced feelings of loneliness during lockdowns [57,58]. Loneliness among young adults is strongly associated with psychiatric disorders and psychosocial risk factors such as depressive symptoms, suicidal ideation, social anxiety, alcoholism, violent behavior, and impulsivity [58]. Loneliness during COVID-19 has also enhanced the anxiety and risk perception of young adults in Poland [59] and was associated with depression and decreased life satisfaction among Middle Eastern adolescents [60]. Loneliness is defined by emptiness, a loss of control, rejection, worthlessness, and personal threat as well as the absence of rewarding social ties—chronic or transient [61]. Students in this study expressed these emotions. The lockdown in Indonesia entailed travel restrictions, the inability to visit friends and relatives, and, for many people, abstaining from employment and leaving the house except in extraordinary circumstances or for immediate, everyday life needs [62].

Apart from feeling lonely and disconnected, the youth experienced pandemic isolation-related sentiments of brokenness, frustration, confusion, and anger. These emotions have been almost universally reported by people worldwide throughout the COVID-19 pandemic. Researchers have recounted distressing emotional experiences during the outbreak across nations and contexts including emptiness, pain, anxiety, fear, depression, rage, worry, and irritation, as reported in Canada [63]; Ireland [64]; the United States [65]; Turkey [66]; and Indonesia [67]. Adolescents and emerging adults are a particularly vulnerable population during the COVID-19 crisis [68]. Depression and anxiety disorders are among the most prevalent mental disorders in children and adolescents [69], resulting in severe functional impairment [70] and an increased risk of suicide [71] During the COVID-19 pandemic, such reported cases have been approximately double the average rate [72].

The youth in this study also reported experiencing conflicting emotions including at times, abrupt mood swings. Two studies conducted in Italy and China found that during the lockdown, many children and adolescents failed to adjust to home learning, resulting in anxiety and mood swings [73] whereas a study in Spain discovered that many women reported suffering mood swings while males reported them at a substantially lower rate. This tendency of a more dramatic increase in mood swings also occurred among younger people [74,75]. In research undertaken in LMICs, conflicting feelings resulting from COVID-19 lockdown have not been clearly described. This could be due to a lack of research or because open expression of emotions is frowned upon in certain cultures. The ability to express oneself emotionally is influenced by one’s cultural background, most notably religion and ethics [75]. In Indonesia, for instance, emotional disclosure is strongly influenced by social, cultural, and religious norms. The Javanese, Indonesia’s largest cultural group, strive for calmness (*ikhlas, tenteram*) in both emotional demeanor and behavior; the Toraja of South Sulawesi maintain social unity by suppressing hostility; and in northern Bali, strong ‘negative’ emotions make one susceptible to a variety of dangerous experiences such as sorcery and spirit possession [76]. Thus, Indonesians are primarily trained to maintain a stable emotional state. However, the youth in this study reported experiencing conflicting emotions and having difficulties in controlling mood swings. The extended pandemic isolation defies normalcy, eliciting a surge of opposing feelings.

Study participants voiced uncertainty about both their current and future lives. At times they were pessimistic, at other times optimistic. Uncertainty about the future can cause emotional disruption and may even contribute to suicidal ideation [72]. A few youths also expressed suicidal ideas in this study. The open-ended, narrative approach may have brought this out in a way that quantitative studies have previously missed. Three participants explicitly indicated a wish to end their lives, attributing this to frustration with the current circumstances and fear for the future. We found their courage to expose their most private thoughts surprising, given that they were all Muslims, and Islam strongly condemns murder, whether directed at another person or toward oneself [77]. Suicide is prohibited, and the severity of the sin is so extreme that some Muslims believe that no one may pray for one who has committed suicide [78].

The study participants did not always experience negative emotions and psychological distress; some expressed a sense of purpose and fulfillment, resulting from engaging in productive activities while in lockdown. These positive sentiments emboldened them to proceed with their lives. Sense of purpose has emerged as a critical psychological resource for individuals of all ages to cultivate amid the COVID-19 pandemic [79]. Sense of purpose benefits individuals throughout their lives and is associated with optimism and perseverance [80]. Several of the youth in the study experienced lockdown in their hometowns, often small rural villages. Although some reported struggling to adapt to village life, others found purpose and fulfillment through volunteer work such as assisting parents in the rice fields and engaging in activities that promoted community [81]. Adopting attitudes of purposeful action thus acted as a resource for coping during the lockdown.

In relation to purpose and meaning was the importance of the students’ faith in coping with the lockdown. Spirituality has been characterized as a dynamic connection to oneself, others, nature, or God in meaning-making frameworks [82]. According to researchers, spiritual viewpoints have been linked to increased tolerance for psychological and physical stress, good aging, and a greater capacity to cope with significant diseases and solitude, according to researchers [83] Additionally, spirituality has been shown to have a beneficial effect on one’s quality of life, levels of happiness, and adaptation [82]. Muslims rely on their faith to recuperate during times of adversity, and their religious beliefs and practices aid them in analyzing a situation and seeking solutions [84]. Islamic religiosity can be a powerful coping mechanism for Muslims against anxiety or depression, which is also prevalent during this pandemic [85].

### Implications for Policy and Practice

Children and adolescents are particularly vulnerable to the effects of continuous stressors during developmental stages, hence their mental health during and after the pandemic requires considerable attention from stakeholders [86]. Efforts should be taken to improve support systems during the pandemic to mitigate the negative impacts of the lockdown on young people’s mental and psychological quality of life. In the current study, some youth were able to turn the difficult moments of COVID-19 lockdown into opportunities by drawing strength from personal resources, family, friends, and supportive communities. Understanding how adolescents perceive pandemic isolation and their well-being can assist psychologists, counselors, parents, and teachers/lecturers to help youth who are experiencing psychological distress.

Drawing on the findings and related studies, we recommend several directions for adolescents, their families, and schools to promote thriving in the face of extended lockdowns. First is to empower young people to use the COVID-19 outbreak as a catapult for redefining and acquiring a deeper sense of purpose in their lives. Jans-Beken [87] refers to this attitude as “mature gratitude” and views it as a method for coping with the threat associated with adversity or times of crisis. She indicates that building an attitude of mature gratitude through acts of kindness, expressing thanks for life and God, and loving the small things in life can support individuals in coping with COVID-19 confusion and developing lifelong resilience for the future.

It is critical to equip adolescents with the ability to persevere in the face of adversity and to aid them in building appropriate coping skills. Positive coping mechanisms and personal resilience have a vital role in overcoming lockdown-induced stress and other pandemic-related mental and psychological health consequences. Personal resilience is required for a successful recovery from extreme or difficult circumstances [88], whereas coping skills aid in resolving or expediting the resolution of problems [89]. Individuals must possess sufficient personal resilience and coping qualities to deal with the pandemic’s negative impacts and maintain their mental health [90].

Youth should be encouraged to form social connections with their peers, families, and communities. Social connections contribute significantly to resilience by acting as a buffer against harmful physical and mental health consequences, particularly during moments of crisis [91]. Families can use stay-at-home restrictions to spend more time together to fortify affective relationships between family members. Pandemic isolation provides an opportunity to strengthen family bonds, which may have a beneficial effect on a family’s mental health [92]. A family connection provides youth with a sense of security during stressful times [93]. Furthermore, several participants in this study demonstrated how their connection with their communities and peers helped them find purpose and fulfillment, which enabled them to remain productive and optimistic during the pandemic.

To foster a more inclusive and welcoming family environment, parents should check in with their children and assist them in overcoming anxiety and depression by generating discussion about the stress of social isolation, which can result in help-seeking behavior. Parents should also monitor their children’s mood and conduct; knowing that this is a stressful period, youth may be more sensitive to anxiety and depressive symptoms. This can result in earlier discovery and access to services [92]. Parents should talk with their children about the predicament, provide accurate and age-appropriate facts, and attempt to understand their children’s emotions, anxieties, and disappointments over the losses caused by the pandemic. In Indonesia, more young people have decided to marry early to escape pandemic depression caused by excessive amounts of schoolwork and domestic responsibilities, and a lack of parental support. These factors together have driven more youth to seek support in romantic relationships, which has resulted in increased rates of marriage [16].

This study contributes to the growing body of literature on the psychological effects of COVID-19 isolation on youth through the study of personal, first-hand narratives. In so doing, the study addresses an important gap in previous research on the subject, which has relied mostly on statistical or secondary data. Nevertheless, important study limitations should be noted. Despite the size of the study sample, a convenience sampling approach was used to collect the data, targeting youth enrolled in higher education, specifically students from a single university faculty. Convenience samples typically comprise a small proportion of underrepresented sociodemographic subgroups that contribute a small amount of variation to the sample; the study’s findings are recognized as generalizable only to the sample analyzed. The study was carried out at one university in an urban center, Jakarta. While some students lived in remote areas, almost 95% of the study sample were from Java Island. Moreover, the age distribution of the youth was less varied; the majority were 19 years old. Furthermore, the sample in the study was not representative of the larger adolescent population; only 10.36% of adolescents in Indonesia completed their education up to the level of college. Inferences cannot be made about adolescent well-being to other youth populations in the country, as the experiences of adolescents from other age groups, and educational and geographic backgrounds may be quite different from the current study sample. 

## 5. Conclusions

In line with several recent studies on adolescent mental well-being during COVID-19, the students in the present study experienced a plethora of psychological responses to the extended lockdown including frustration, despair, anger, and alienation. Clearly, psychological stress and changes in individual and family routines brought about by home confinement have a detrimental impact on the health and well-being of youth. Economic loss and uncertainty including direct loss of income by the students and/or their families was a major contributing factor to the students’ psychological and emotional well-being. While a common experience for many people around the world during the COVID-19 pandemic, few studies have captured adolescent accounts of how economic loss has impacted their psychological well-being. Such economic losses can be devastating and deeply affect the students’ ability to function while attempting to continue their studies. Despite the many challenges the students faced, their stories also unveiled important leverage points including faith and family relationships, which can be harnessed as a platform for mitigating the repercussions of the lockdown. For an LMIC like Indonesia, all available personal and ecological assets should be mobilized to increase the quality of the country’s productive age population. Doing so will enable its youth population to serve as a resource for the country’s advancement.

## Figures and Tables

**Figure 1 ijerph-18-10489-f001:**
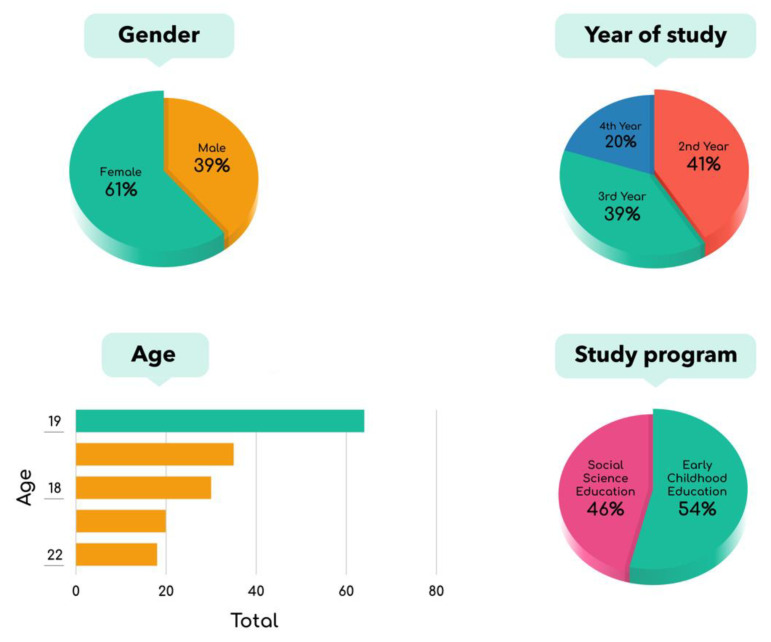
Demographic profile of the study participants.

**Figure 2 ijerph-18-10489-f002:**
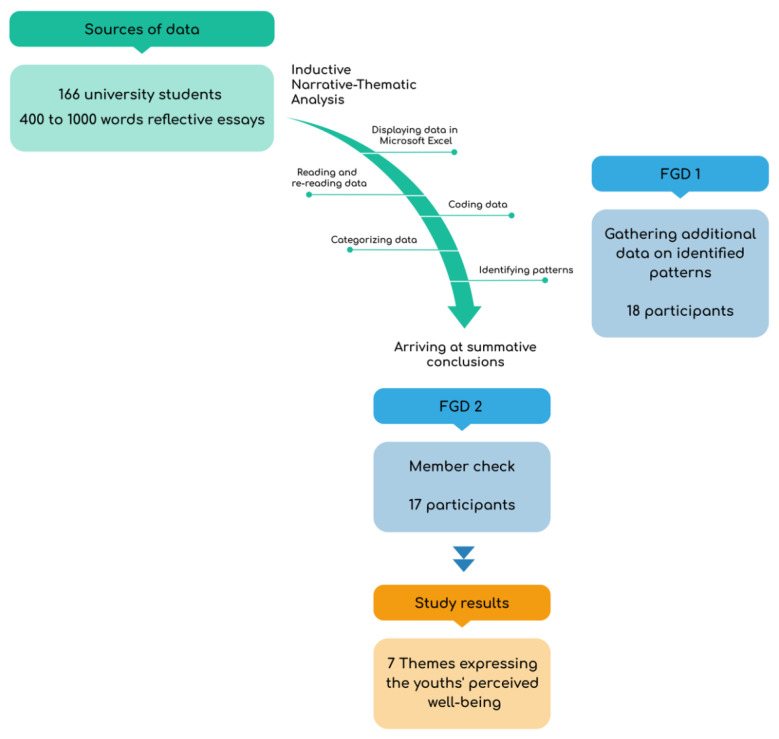
Research procedures.

**Figure 3 ijerph-18-10489-f003:**
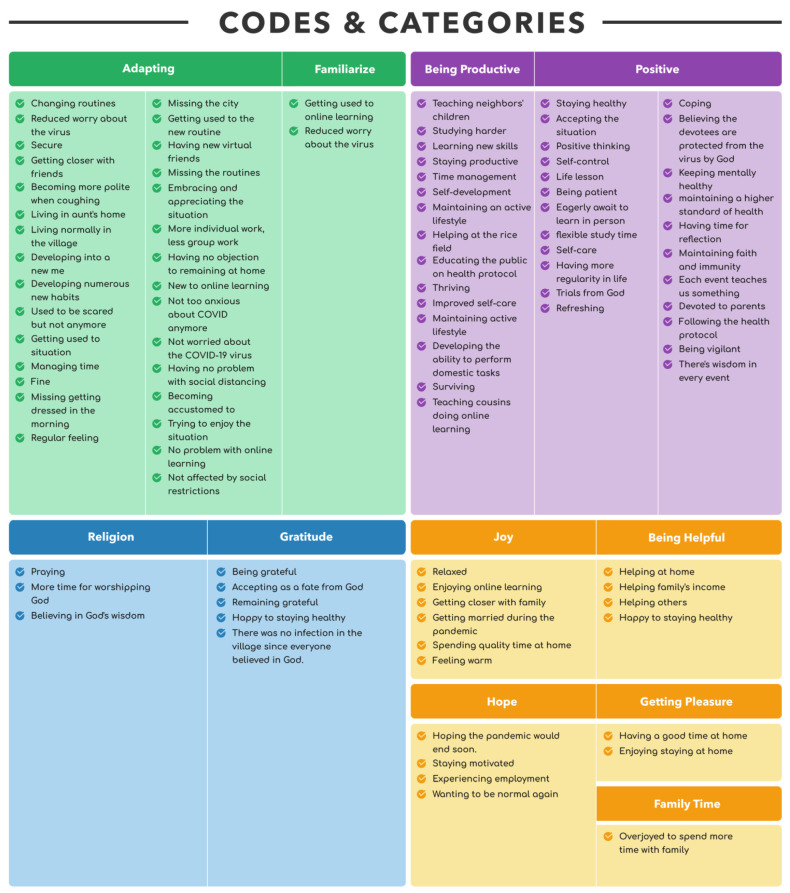
Positive codes and categories.

**Figure 4 ijerph-18-10489-f004:**
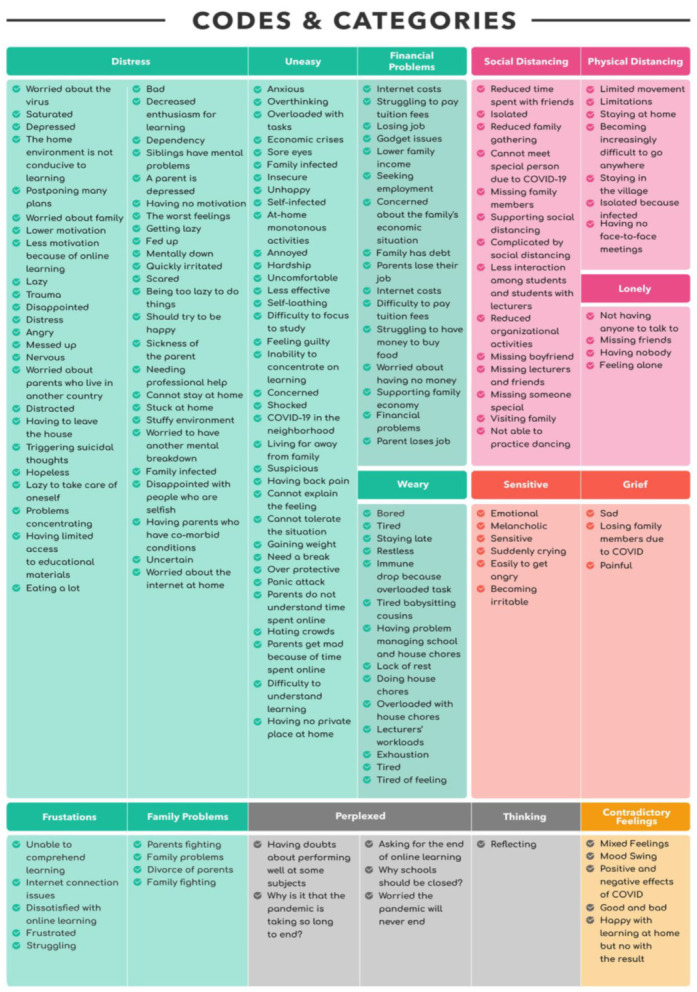
Negative codes and categories.

**Figure 5 ijerph-18-10489-f005:**
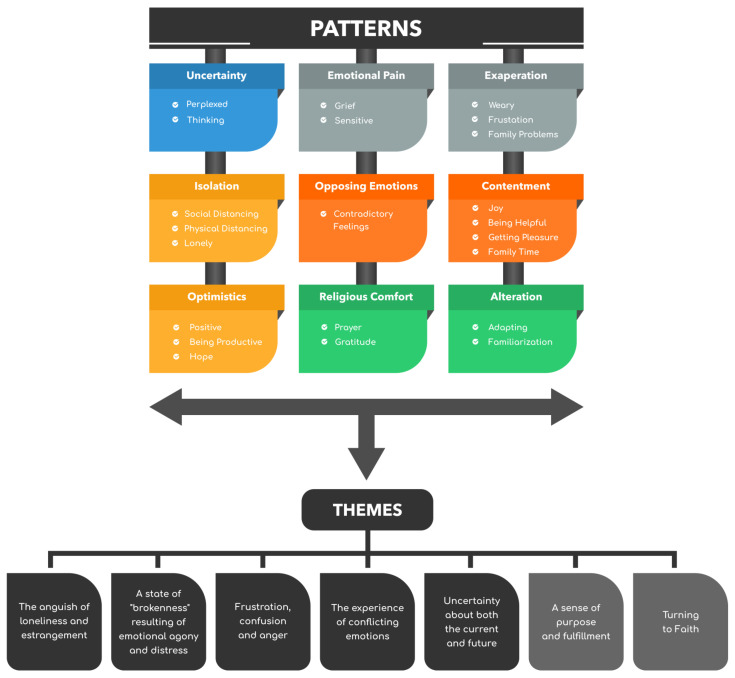
Patterns and themes.

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
