# Peer review of "Perceived Consequences of Extended Social Isolation on Mental Well-Being: Narratives from Indonesian University Students during the COVID-19 Pandemic"

_ijerph, 2021, doi:10.3390/ijerph181910489_

Round 1
Reviewer 1 Report
The article raises an interesting objective, mainly because it focuses on a country with a weak socioeconomic context. The methodological approach is also interesting and successful because it allows us to delve into the objective of the study.
Taking into account that the pandemic is recent, the theoretical framework is complete and representative of the research related to the subject of the article.
On the other hand, I consider that the results are complete and the conclusions are consistent with the research approach.
Despite the positive aspects, I think there are a few elements that could improve: firstly, it starts from the premise that young university students are a population with a higher rate of mental health problems, eating disorders, anxiety, and substance abuse … But they only cite research on this claim. Given the importance of the subject, I believe that they should better justify what is said, they only name: Browning, M. H. E. M .; Larson, L. R .; Sharaievska, I .; Rigolon, A .; McAnirlin, O .; Mullenbach, L .; Cloutier, S .; Vu, T. 773 M .; Thomsen, J .; Reigner, N. Psychological Impacts from COVID-19 among University Students: Risk Factors 774 across Seven States in the United States. PloS one 2021, 16 (1), e0245327
On the other hand, it would be appropriate to better clarify (perhaps a table) the categories and subcategories used, this would facilitate reading.
Finally, congratulate the authors for the work done.
Reviewer 2 Report
This study is meaningful and will be useful in the relevant field particularly during and after the Pandemic situations in that the findings of this study can contribute to the field on the adverse psychological effects of COVID-19 isolation on youth. In particular, I believe that the methodology section is very strong and clear in terms of qualitative narrative characteristics of the study. The study are very rigorous in that it included essays, FGDs, and member checks to collect data, analyze, and triangulate the findings.
However, there are some stylistic errors (grammar, repetition of the same word, and so on) throughout the paper; thus, it would be required to check the manuscript thoroughly.
Here are a few questions.
The Current Study: The authors introduced previous studies, so would it be studies instead of study?
Methods: Participants – It would be better to use a table to explain the participants so that readers can understand the contexts and participants of the study better.
The authors mentioned that they used non-biased and non-leading questions by asking participants to express their feelings in the essay and encouraged additional comments during the FGD sessions. Thus, I guess it would be better to show the non-biased and non-leading questions as an attachment or some samples in the text.
Thank you.
Reviewer 3 Report
The subject is of maximum interest, like everything derived from the COVID-19 pandemic, as well as what refers to the mental health of adolescents as a result of the pandemic (adolescents; mental health ...).
Furthermore, there are very few studies and research on the situation of children and young people in low- and middle-income countries (LMICs), which are taking place in rich countries.
“While studies are still scant compared to those from wealthier nations, COVID-19's 68 impact on children and youth in low- and middle-income countries (LMICs) may be sub-69 stantially more severe.”
Therefore, I do not question the interest and timeliness of the topic of the article.
The authors point out that this research is part of a larger study on the resilience and well-being of college students, of which numerous articles will have or will have been published, and I get the impression that in that case it has been forced, or used to get an article, which takes advantage of the method and instruments.
The conclusions are poor, very basic, not very systematized…: “The study contributes to the growing body of literature on the adverse psychological effects of COVID-19 isolation in young people; It is a study different from the usual ones (based on statistical or secondary data); psychological stress from confinement has a detrimental impact on the health and well-being of young people; a country like Indonesia, with low GDP, must make every effort to increase quality and advance the youth population…”
In my opinion, the authors are aware of the shortcomings of the article, and acknowledge it: "Important study limitations should be noted".
Therefore I suggest the complete review of the article. Regarding results and conclusions, as well as shortcomings at a methodological level (despite the fact that they explain the process followed): method followed, choice of sample ...
Reviewer 4 Report
1.-The narrative inquiry may not be the most appropriate for these types of studies.
No statistical results are presented that allow reaching scientific conclusions. It would be appropriate to incorporate percentages
of the different results obtained with the treatment of the questionnaires
2.-The investigation, as the text itself recognizes, is part of a larger study, perhaps for that reason tables and graphs are not included to clarify the results obtained, however we miss it.
3.- 130 bibliographic citations seem excessive to us for an article of this length. Please reduce their number so that only bibliographic references with scientific evidence that have allowed them to carry out their research are included.
4.- The conclusions lack comparisons with those of other studies already published. Likewise, a study of this magnitude should explain in more detail the limitations it has found.
Finally, in this pandemic situation, which is still in force, is missing what contributions the authors estimate or recommend to improve the population under study.
Round 2
Reviewer 3 Report
The article, from my point of view, has improved enormously. It is more coherent and better explained.
Especially in the section referring to the method (completed with charts) and conclusions (convenient section modified).
The authors have also reviewed other aspects indicated in the initial evaluation (limitations, discussion,...).
Other modifications, changes and additions, which must have been 'suggested' by other evaluators, I also consider as appropriate and beneficial.
Reviewer 4 Report
Thanks for the clarifications and corrections made. Great job